# Nanocrystallization of Anthocyanin Extract from Red-Fleshed Apple ′QN-5′ Improved Its Antioxidant Effect through Enhanced Stability and Activity under Stressful Conditions

**DOI:** 10.3390/molecules24071421

**Published:** 2019-04-11

**Authors:** Xiang Zhang, Heqiang Huo, Xiaohong Sun, Jun Zhu, Hongyi Dai, Yugang Zhang

**Affiliations:** 1College of Horticulture, Qingdao Agricultural University, Qingdao 266109, China; 18306391375@163.com (X.Z.); mingsun9887@163.com (X.S.); junzhu@qau.edu.cn (J.Z.); hydai@qau.edu.cn (H.D.); 2Qingdao Key Laboratory of Genetic Development and Breeding in Horticultural Plants, Qingdao Agricultural University, Qingdao 266109, China; 3Mid-Florida Research and Education Center, University of Florida, Apopka, FL 32703, USA; hhuo@ufl.edu

**Keywords:** oxidative stress, red-fleshed apple, anthocyanin, antioxidant, nanocrystallization

## Abstract

Red-flesh apples are known as functional fruits because of their rich anthocyanin. The anthocyanin content of the red flesh apple cultivar ′QN-5′ we bred can reach 361 mg·kg^−1^ (FW), and showed higher scavenging capacity to DPPH radicals, hydroxyl radicals, and superoxide anion radicals, with scavenging rates of 80.0%, 54.0%, and 43.3%, respectively. We used this particular anthocyanin-rich ′QN-5′ apple as material to examine how nanocrystallization affects the antixodiant effect of anthocyanin. The anthocyanin extract was encapsulated with biocompatible zein to form zein-anthocyanin nanoparticles (ZANPs). Transmission electron microscopy (TEM) scanning showed that ZANPs had a regular spherical shape with an average diameter size of 50–60nm. When the ratio of the zein and the anthocyanin was 1:0.5, the results suggested that the encapsulation efficiency (EE) of the ZANPs could reach as high as 92.8%, and that scavenging rate for DPPH radicals was increased from 87.1% to 97.2% compared to the non-nanocrystallized anthocyanin extract. Interestingly, treatment under alkaline conditions (pH 9.0), high temperature (90 °C), and a storage time of 7 days could decrease the scavenging capacity of the ZANPs for DPPH radicals, but this scavenging capacity loss for ZANPs was significantly lower than that observed in the non-nanocrystallized anthocyanin, suggesting the higher stability of ZANPs is caused by encapsulation. These results would provide a theoretical basis for the application of the anthocyanin in scavenging free radicals under stress conditions.

## 1. Introduction

Anthocyanin is one of flavonoid compounds [1], which had strong antioxidant [2,3,4], anti-cancer [5], and free radical scavenging ability [6]. Red-fleshed apples (*Malus sieversii f. Neidzwetzkyana* (Dieck) Langenf) are precious because their flowers, leaves, and fruits are rich in anthocyanin [7]. The anthocyanin extract of the red-fleshed apple could reduce the oxidative damage caused by reactive oxygen species (ROS) on pig ovarian granulosa cells [8]. ROS are a group of oxygen-containing substances with active chemical properties and a strong oxidation ability, which mainly includes superoxide anion radicals, hydrogen peroxide, hydroxyl radicals, lipid free radicals, etc. [9]. Excessive production of free radicals and lack of antioxidants leads to oxidative stress (OS) in organisms, which plays a key role in maintaining the homeostasis of organisms. The antioxidant properties of fruit anthocyanin in scavenging free radicals has been scientifically proved [10]. Anthocyanin can interact with free radicals via phenolic hydroxyl groups to form semiquinone free radicals, which eliminates excess free radicals and terminates their chain reaction. In return, this process further inhibits the excessive production of ROS, and maintains a dynamic balance by regulating the production and consumption of free radicals in the body, in order to reduce the damage caused by OS [11]. In addition, anthocyanin acts as a strong antioxidant to protect visual acuity by protecting photoreceptor cells in the retina from damages caused by OS [12].

Nanocrystallization of anthocyanin is an encapsulating process with macromolecule polymers under various physical or chemical treatments used to form nanoparticles with uniform dispersion. In nature, anthocyanin is extremely unstable, due to degradation caused by external factors such as pH, temperature, and other factors [13], and degradation of anthocyanin might affect its ability to scavenge free radicals. For example, procyanidins are sensitive to ultraviolet and other stresses, but their degradation under these stressful conditions could be substantially alleviated by protein encapsulation of procyanidins, indicating the role of nanocrystallization in improving the stability of anthocyanin.

The free radical scavenging capacity of nanocrystallized proanthocyanidins is significantly stronger than that of pure proanthocyanidins under stressful conditions [14]. Arroyo-Maya et al. found that protein-encapsulated anthocyanin could improve its stability during storage [15]. Liu et al. found that zein was an excellent material for highly unsaturated fatty acids to use to maintain their antioxidant stability [16]. 

In this study, zein was used for the encapsulation of high concentration anthocyanin isolated from red-fleshed apple ′QN-5′ variety that could meet encapsulation requirement, and they were formed into zein-anthocyanin extract hybrid nanoparticle (ZANPs). Our results showed that the ZANPs exhibited higher stability under alkaline conditions (pH 9.0), high temperature (90 °C), and a storage time of 7 days. These results would provide a deep insight into understanding the role of nanocrystallization in enhancing the antioxidant ability of anthocyanin, and a fundamental rational for the application of encapsulated anthocyanin to alleviate cell damages by OS.

## 2. Results

### 2.1. Contents of Anthocyanin and Total Phenols in the Peel and the Flesh of the Red-fleshed Apple ′QN-5′

The ′QN-5′ fruits had red peels and flesh (Figure 1A,B), and the extracts from the peel (Figure 1D) or the flesh (Figure 1C) were used for determining the content of the anthocyanin and total phenols with a pH differential method. The anthocyanin content of the flesh was 1.4-fold higher than that of the peel, with 361.0 mg·kg^−1^ (FW) in the flesh and 257.8 mg·kg^−1^ (FW) in the peel (Figure 1E). However, the total phenol content in the peel was 2724.6 mg·kg^−1^ (FW), which was 2.3-fold higher than that of the flesh at 1198.8 mg·kg^−1^ (FW) (Figure 1F). The anthocyanin content of the peel account for only 9.3% of its total phenols, whereas 30.1% of the total phenols in the flesh was anthocyanin. The anthocyanin extract of pulp was used to prepare ZANPs because of its high anthocyanin content.

### 2.2. Antioxidant Activity of the Anthocyanin Extract from the Red-fleshed Apple ′QN-5′ in vitro

The anthocyanin extracts of the red-fleshed apple ′QN-5′ had a stronger ability to scavenge DPPH free radicals, hydroxyl free radicals, and superoxide anion free radicals, compared with the Vitamin C (VC) (Figure 2A–C). However, the ′QN-5′ extracts had higher antioxidant activity to scavenge DPPH free radicals than to remove hydroxyl free radicals and superoxide anion free radicals (Figure 2A–C). The highest scavenging rates of the extracts from the peel and the flesh for DPPH free radicals, hydroxyl free radicals, and superoxide anion free radicals were 91.1%, 54.1%, and 43.3%, respectively, at different concentrations of 10 mg·kg^−1^, 30 mg·kg^−1^, and 50 mg·kg^−1^ (Figure 2A–C). There was no significant difference in DPPH free radical scavenging rates of the extracts from the peel and the flesh at three concentrations. At low anthocyanin concentration (0–10 mg·kg^−1^), the DPPH scavenging rate increased significantly with the increase of anthocyanin concentration (Appendix A). The scavenging rates of the flesh extract for hydroxyl free radicals and superoxide anion free radicals were higher than those of the peel extract at the concentrations of 30 mg·kg^−1^ and 50 mg·kg^−1^. At the concentration of 10 mg·kg^−1^, the scavenging rates for hydroxyl and superoxide anions of the flesh extract were lower than those of the peel extract, which might be due to the higher dilution ratio of the flesh extract. The flesh extract was diluted by 35 times from the original extract to 10 mg·kg^−1^, while the peel extract was diluted only by 25 times (Figure 2D) to the same concentration.

### 2.3. Characteristics and Antioxidant Properties of Zein-anthocyanin Hybrid Nanoparticles (ZANPs)

The anthocyanin extract of the red-fleshed apple was mixed with the nanocrystallized zein solution in different proportions (Figure 3A). The particle size and PDI (polymerization dispersion index) of ZANPs increased with the increase of anthocyanin amount, but reached their maximum values when the zein-to-anthocyanin ratio was 1:1.0. The scavenging capacity of the ZANPs with different proportions of anthocyanin was also different for DPPH free radicals (Figure 3B). For example, the scavenging rates of the ZANPs with a zein-to-anthocyanin ratio of 1:0 and 1:0.1 were less than 20% (Figure 3B). By contrast, the DPPH free radical scavenging rates of the ZANPs with zein-to-anthocyanin ratio of 1:0.5 and 1:1.0 were 97.2% and 97.5%, respectively. The anthocyanin encapsulation efficiency significantly varied with various zein-to-anthocyanin ratios (1:0.1, 1:0.5, 1:1.0). The highest encapsulation efficiencies of 92.8% and 74.8% were observed at the zein-to-anthocyanin ratios of 1:0.5 and 1:1.0, respectively. Given that the DPPH free radical scavenging rate and encapsulation efficiency were the highest when the zein-to-anthocyanin ratio was 1:0.5, the ZANPs at this ratio were used for examining their properties. The zein alone without anthocyanin extract in the ethanol solution greatly agglomerated, as shown in Figure 3C. After nanocrystallization, spherical zein-nanoparticles with mean particle sizes ranging from 20 nm to 30 nm and PDI ranging from 0.2 to 0.3 were formed, showing a homogeneous size distribution (Figure 3D,E). When the anthocyanin was encapsulated in the nanoparticles to form ZANPs, the average particle sizes ranged from 50 nm to 60 nm, which was 1.6- to 3.0-fold larger than that of pure zein nanoparticles. The shape of the ZANPs was spherical or quasi-spherical (Figure 3F). 

### 2.4. Stability and Antioxidant Activity of the ZANPs In Vitro

#### 2.4.1. Stability and Antioxidant Activity of the ZANPs under Alkaline Conditions In Vitro

As the anthocyanin was encapsulated in the nanoparticles to form the ZANPs, the antioxidant activity of the ZANPs was significantly enhanced. The DPPH free radical scavenging rate was increased from 87.1% to 97.2%, which was 8.6 and 1.5-fold higher than those of the zein (11.3%) and VC (63.7%), respectively (Figure 4A). After alkaline treatment (pH 9.0) for 10 min, the scavenging rates of the zein, the ZANPs, the anthocyanin extract, and VC for the DPPH free radicals decreased to various degrees (Figure 4B). The scavenging rate of the anthocyanin extract decreased from 87.1% to 0.7%, and the decrease rate was 99.19%. After nanocrystallization, the percentage of the scavenging rate of the anthocyanin decreased from 97.2% to 7.5%, and the decrease rate was 92.28%, which was significantly lower than that of non-nanocrystallized anthocyanin.

#### 2.4.2. Stability and Antioxidant Activity of the ZANPs under High Temperature in vitro

After alkaline treatment at 90 °C for 30 min, the scavenging rates of the zein, the ZANPs, the anthocyanin extract, and the VC on DPPH free radicals decreased to various degrees. The scavenging rate of VC decreased from 63.7% to 35.8% (Figure 5A). The scavenging rate of the zein decreased from 11.3% to 6.5%. The scavenging rate of the anthocyanin extract decreased from 87.1% to 82.8% and the decrease rate was 4.93% (Figure 5B). The scavenging rate of ZANPs decreased from 97.2% to 96.7%, and the decrease rate was only 0.51%, which was significantly lower than that of the non-nanocrystallized anthocyanin.

#### 2.4.3. Stability and Antioxidant Activity of the ZANPs after 7 days of Storage In Vitro

The scavenging rate of four active substances on DPPH free radicals decreased significantly after 7 days’ storage at 4 °C (Figure 6A). The highest decline in scavenging rate was observed in VC, from 63.7% to 6.3%. The non-nanocrystallized anthocyanin exhibited a decrease in scavenging capacity from 87.1% to 73.5%, with a decrease rate of 15.6% (Figure 6B). By contrast, the scavenging rate of ZANPs decreased from 97.2% to 88.5%, resulting in an 8.9%, decrease rate that was significantly lower than that of non-nanocrystallized anthocyanin.

## 3. Discussion

In this paper, the red-fleshed apple variety ′QN-5′ had higher anthocyanin content, higher scavenging capacity of the DPPH free radicals, hydroxyl radicals, and superoxide anion free radicals, and its scavenging capacity was significantly stronger than that of VC. Although the total phenol content in the peel was higher than that in the flesh, the anthocyanin content in the peel was lower than that in the flesh. So, the proportion of anthocyanin in total phenols in the flesh was higher than that in the peel, and anthocyanin extracted from the flesh was used as material for subsequent experiment. 

Oxidative stress (OS) occurred when the balance between ROS generation and antioxidant activity was disturbed, or the rate of formation of free radicals exceeded the antioxidant capacity of the system [17]. Therefore, as a strong antioxidant, the anthocyanin extract of the red-fleshed apple had a good ability to scavenge free radicals. By scavenging free radicals, it could reduce the rapid increase of ROS caused by its enrichment, maintain the balance of oxidation and antioxidation levels, and reduce the level of OS caused by the accumulation of free radicals.

Because of degradation of the anthocyanin under stress conditions such as pH and temperature, anthocyanin should be encapsulated in protein to enhance its stability. The zein-anthocyanin hybrid nanoparticles (ZANPs) were successfully prepared by further crosslinking and co-nucleation of anthocyanin molecules in the red-fleshed apple extract with zein as a carrier. Nanoparticles were characterized by small particle size, good dispersion, regular spherical shape, and chemical bonds and molecular forces on the surface. When the anthocyanin molecules were encapsulated into nanoparticles, the uniformly distributed nanoparticles that had relatively large specific surface area and the higher surface free energy between the particles would make the function of the nanomaterials more prominent than that of the raw materials, which would improve their related properties [18]. The average diameter of nanoparticles measured by DLS was larger than that measured by TEM, which might be due to the hydrodynamic diameter measured by DLS, and this result was consistent with that of Sarmphim et al. [19].

After nanocrystallization of the anthocyanin, the stability and free radical scavenging ability were enhanced by strong alkaline conditions, high temperature treatment, and storage, which promoted the inhibition of OS. Three stress conditions, strong alkaline (pH 9.0), high temperature (90 °C), and 7 days′ storage, had strong inhibitory effects on the activity of active substances, and weakened the scavenging ability of active substances on the DPPH free radicals. Also, the weakening effect of strong alkaline conditions was significantly stronger than that of high temperature and long-term storage conditions. Nanocrystallization of the anthocyanin could alleviate the weakening effect of antioxidant activity of active substances under stress conditions, which might be due to the encapsulation of the anthocyanin into zein, which acted as a shell to protect the anthocyanin and prevented its premature degradation [20]. 

However, the mechanism of scavenging free radicals by the ZANPs and the chemical bond and interaction between zein and anthocyanin in hybrid nanoparticles remains unclear. The mechanism that allows nanocrystallization to enhance the antioxidant activity of the anthocyanin and its stability under stress conditions needs to be further explored.

In conclusion, the anthocyanin extract of the red-fleshed apple cultivar ′QN-5′ could be used as a strong antioxidant. Nanocrystallization could improve its antioxidant effect through enhancing stability and activity under stressful conditions, and reduce the OS level caused by free radical accumulation.

## 4. Material and Methods

### 4.1. Materials

The red-fleshed apple ′QN-5′ was a new cultivar bred by our team and was grafted on *Malus Robusta* for 5 years at the experimental farm of Qingdao Agricultural University (Qingdao, China). The fruits were collected in late August and brought back to the laboratory with an ice box. The peel and the flesh of the fruits were separated and ground into powder under liquid nitrogen. Extraction of anthocyanin from the red-fleshed apples was carried out with absolute ethanol at the ratio of 1:10 (fruit tissues to extraction buffer) under dark conditions for 15 h. The extract was subsequently filtered with a 0.45 μm membrane and stored at −4 °C. Zein was purchased from Shanghai Solarbio Science and Technology Co., Ltd (Shanghai, China). Folin-ciocalteu was purchased from Beijing Solarbio Science and Technology Co., Ltd (Beijing, China).

### 4.2. Determination of Total Anthocyanin Content

Total anthocyanin contents were determined using a pH differential spectroscopic method [21]. 1 mL of the anthocyanin extract was added into 9 mL of potassium chloride buffer (0.025 mol/L, pH 1.0) and 9 mL of sodium acetate buffer (0.4 mol/L, pH 4.5), respectively. The mixture of each sample was incubated for 60 min at room temperature, and the absorbance of each sample was measured at 510 and 700 nm using a spectrophotometer. Absorbance (A) of each sample was calculated as follows: A = (A510 − A700)pH 1.0 − (A510 − A700)pH 4.5; Total anthocyanin content (mg/kg FW) = A × MW × DF/(ε × W). Where MW (449.2) is the molecular weight of cyanidin-3-glucoside (predominant anthocyanin in the extracts), DF is the dilution factor, ε (26,900) was the molar absorptivity of cyanidin-3-glucoside, and W is the fresh weight of each sample.

### 4.3. Determination of Total Phenol Content

The Folin-Ciocalteu method was used to determine total phenol content, as modified and described in the reference material [22]. The gallic acid standard solution with a certain concentration gradient was prepared by diluting a series of different volume of standard mother liquor with distilled water. In a 5 mL volume system, gallic acid solution with different concentrations was added in turn and 10% sodium carbonate solution was added in 2–5 mins later. The absorbance of these solutions was determined at 765 nm after incubation at 50 °C for 1 h. According to the concentration gradient of gallic acid and the measured absorbance, the corresponding standard curve was drawn. Each sample was diluted 50 times, and the absorbance of each diluted sample was determined as described for standard solutions. Total phenol content of each sample was calculated as follows: total phenol content (GAE mg/kg FW) = C × V × N/m. Where C is the concentration calculated from standard curve, V is final volume of sample solution, N is dilution multiple, and m is the weight of sample.

### 4.4. Scavenging Capacity of the DPPH Free Radicals

The scavenging method of DPPH free radicals was slightly modified with reference to the method of He [23].The mixture of 2 mL extract and 2 mL DPPH (0.2 mmol/L) free radicals was recorded as Ai. The mixture of 2 mL extract and 2 mL ethanol was recorded as Aj. The mixture of 2 mL DPPH free radicals and 2 mL ethanol was recorded as Ac. The three groups reacted for 30 min in the dark. The absorbance of each group was determined at 517 nm. The scavenging rate was calculated as follows: Scavenging rate (%) = [1 – (Ai−Aj)/Ac] × 100%.

### 4.5. Scavenging Capacity of ·OH^−^ Free Radicals

The modified method of phenanthrene-Fe^2+^ oxidation [24] was used to determine the scavenging rate of ·OH^−^ free radicals. There were three groups named the injured group, uninjured group, and sample group. 1.5 mLphenanthrene solution (5 mmol/L) was added to 9 mL PBS buffer (0.01 M, pH 7.4), and 1 mL FeSO_4_ solution (7.5 mmol/L) was added into the mixture. Then, 2.5 mL distilled water was introduced into the injured and uninjured groups when 2.5 mL extract of red-fleshed apple was added into the sample group. 1 mL 1% H_2_O_2_ solution was then introduced into the injured group and sample group when 1 mL distilled water was added into uninjured group. Each absorbance was determined at 536 nm after three groups were set at 37 °C for 1 h. The scavenging rate of ·OH^−^ free radicals was calculated as follows: Scavenging rate (%) = (As − Ai)/(Au − Ai) × 100%. Where As was the absorbance of sample group; Ai was the absorbance of injured group; Au was the absorbance of uninjured group.

### 4.6. Scavenging Rate of O_2_^−^ Free Radicals

Pyrogallol autoxidation method [25] was used to determine scavenging rate of red flesh apple extract on O_2_^−^ free radicals. There were two groups called blank group and sample group. Taking 0.1 mL distilled water and 0.1 mL extract into blank group and sample group after 4.5 mL Tris-HCL buffer (pH 8.0) was added into two groups. After two groups being incubated at 25 °C, 0.4 mL pyrogallol (2.5 mmol/L) was added into two groups respectively. Two drops of HCL (8.0 mol/L) was added into groups to terminate the reaction. The absorbance was determined at 325 nm. Scavenging rate of O_2_^−^ free radicals was calculated as follows: Scavenging rate (%) = (Ab − As)/Ab × 100%. Where Ab was the absorbance of blank group and As was the absorbance of sample group.

### 4.7. Preparation of Zein-Anthocyanin Extract Hybrid Nanoparticle (ZANPs)

Zein powder (0.01 g) was dissolved in 100 mL 85 wt% ethanol solution (0.1 mg/mL). The pH of the protein solution was adjusted to 11 with 2 M NaOH solution. Sealed with sealing film, the protein solution was incubated at 98 °C for 40 min. The zein suspension was slowly changed to the original volume with absolute ethanol. The suspension was placed in an ultrasonic cleaner for 3 min. Then, the protein solution was incubated under magnetic agitation at 30 °C for 3 min. 100 mL of anthocyanin extract that came from flesh (0.01, 0.05, and 0.1 mg/mL) were added dropwise into 100 mL zein ethanol solution, while being mixed under continuous stirring (600 r/min) at 50 °C. After adding the anthocyanin extract, the mixture was further incubated at 30 ℃ for 3 h. 1/8 volume of 3 mM CaCl_2_ solution was dropped and the mixture continued to be stirred for 3 h in dark conditions. Then, the formed particle suspension was cooled down to room temperature (25 °C) for determining particle size and other physical properties. The suspension was centrifuged at 10,000 g for 10 min to obtain dry particles. The sediment was washed three times with water and lyophilized for 36 h. The supernatant was collected to calculate encapsulation efficiency (EE). The anthocyanin content was determined by UV-visible spectrophotometric method [15]. 1 mL of supernatant was added into 9 mL of DMSO. The absorbance was determined at 520 nm, where the absorbance was determined in the absorption spectrum. The anthocyanin content was calculated by substituting absorbance in Formula (4.2).

The EE of anthocyanin was calculated as follows:(1)$$\mathrm{EE}\left(\%\right)=\frac{\left(total\;content\;of\;anthocyanin\right)-(content\;of\;anthocyanin\;in\;supernatant)}{total\;content\;of\;anthocyanin}$$

### 4.8. Size and Polydispersity Index of the ZANPs

The mean particle diameter and polymerization dispersion index (PDI) of the ZANPs were determined by dynamic light scattering technique using a Zetasizer Nano ZS90 (Malvern Instruments, Malvern, UK). The suspension was diluted 30 times before determination. 

### 4.9. Transmission Electron Microscopy (TEM) Analysis

The morphology and size of the ZANPs were measured with a Hitachi 7650 transmission electron microscope (TEM) (Tokyo, Japan), at an acceleration voltage of 80 kV. A droplet of the ZANP suspension was drop-cast onto a carbon-coated copper grid (400 mesh) and was lyophilized for observation. The range of particle size was calculated by the scale in the TEM image.

### 4.10. In vitro Antioxidant Activity of the ZANPs and Its Stability under Stress Conditions

The DPPH free radicals can be typically used for in vitro antioxidant evaluation of antioxidants [26]. The scavenging rates of the anthocyanin extract and the ZANPs on the DPPH free radicals under alkaline conditions (pH 9, 10 min), high temperature (98 °C, 30 min), and storage (7 days) were determined. The specific methods of scavenging the DPPH free radicals were shown in 4.4.

### 4.11. Statistical Analysis

All the experiments were conducted in triplicate. The data presented as means ± SE. Significant differences among groups were calculated using a Student′s or one-way ANOVA test. Graph-Pad prism5 (San Diego, CA) analysis software and Origin 9.0 software (Northampton, MA, USA) were used to test multiple comparisons and plot charts.

## Figures and Tables

**Figure 1 molecules-24-01421-f001:**
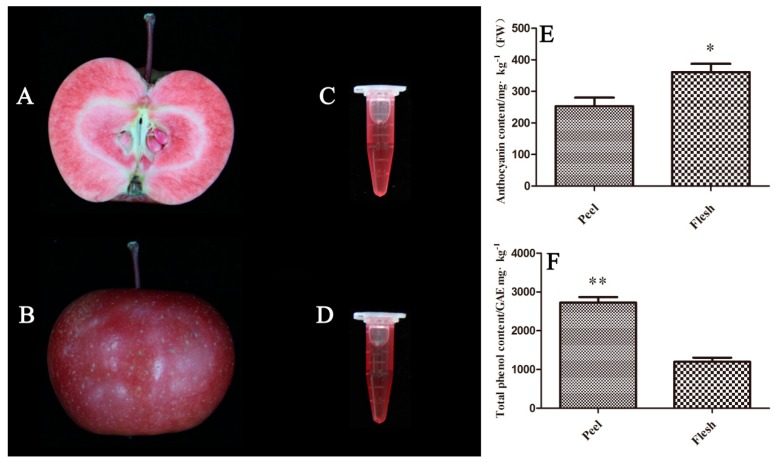
Content of anthocyanin and total phenols in the peel and the flesh of red-fleshed apple variety ′QN-5′. (**A**) Sectioned apple showing red fruit flesh. (**B**) An intact apple showing red peel. (**C**) liquid flesh extract showing red color. (**D**) Liquid peel extract showing red color. (**E**) Anthocyanin content in the peel and flesh. (**F**) Total phenol content in the peel and flesh. * indicated significant difference (*p* < 0.05), ** indicated statistically significant difference (*p* < 0.01).

**Figure 2 molecules-24-01421-f002:**
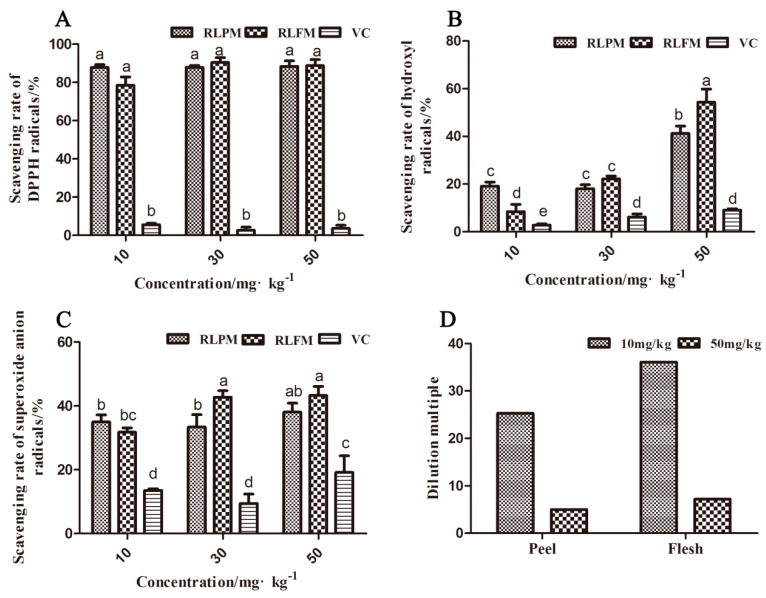
Antioxidant activity of anthocyanin extract from the ′QN-5′ in vitro. (**A**) Scavenging rate of anthocyanin extract on DPPH free radicals. (**B**) Scavenging rate of anthocyanin extract on ·OH^−^ free radicals. (**C**) Scavenging rate of anthocyanin extract on O_2_^−^ free radicals. (**D**) Dilution multiple of extract at corresponding concentration. Within samples, values with different letters (a, ab, b, bc, c, d and e) are significantly different (*p* < 0.05).

**Figure 3 molecules-24-01421-f003:**
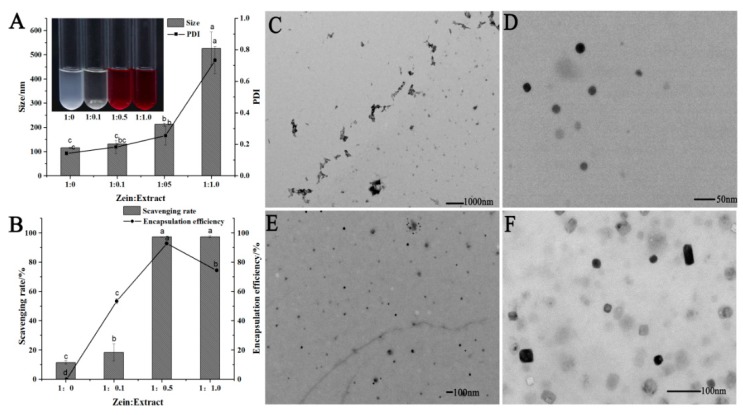
Characterization and morphology of the zein-anthocyanin nanoparticles (ZANPs). (**A**) Size and polymerization dispersion index (PDI) of ZANPs, which were determined by DLS at zein-to-anthocyanin ratios of 1:0, 1:0.1, 1:0.5, and 1:1.0, respectively. (**B**) Scavenging rate for DPPH free radicals and encapsulation efficiency of ZANPs with zein-to-anthocyanin ratios of 1:0.1, 1:0.5, and 1:1.0, respectively. (**C**) TEM (transmission electron microscope) image of zein without anthocyanin extract in the ethanol solution. (**D**) TEM image of zein nanoparticles. (**E**) TEM image of zein nanoparticles distribution. (**F**) TEM image of ZANPs at zein-to-anthocyanin ratios of 1:0.5. Within the same index, values with different letters (a, b, bc, c and d) are significantly different (*p* < 0.05).

**Figure 4 molecules-24-01421-f004:**
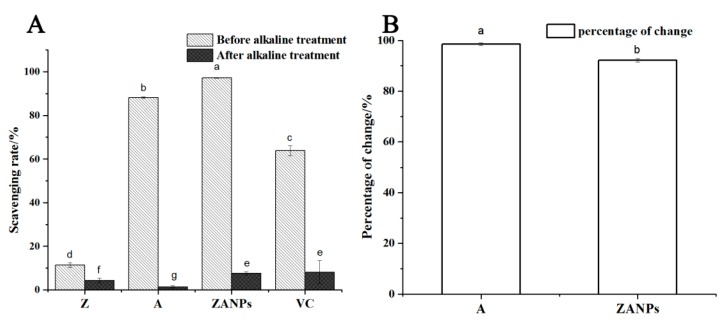
Stability and antioxidant activity of the ZANPs and the anthocyanin extract under alkaline conditions in vitro. (**A**) Scavenging rate of anthocyanin extract and ZANPs on DPPH free radicals under alkaline treatment. (**B**) Percentage change in scavenging rate of A (Anthocyanin extract) and ZANPs. Within samples, values with different letters (a, b, c, d, e, f and g) are significantly different (*p* < 0.05).

**Figure 5 molecules-24-01421-f005:**
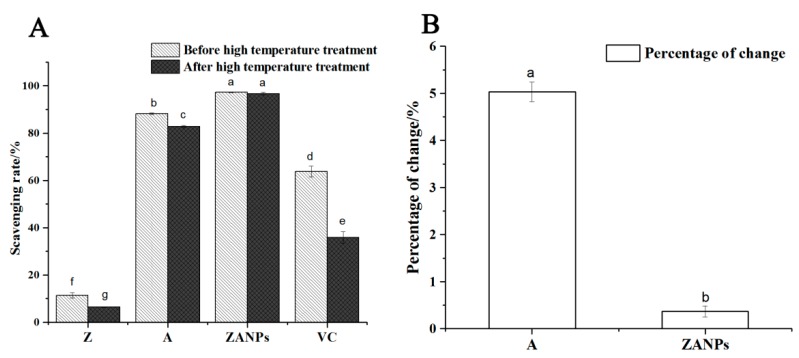
Stability and antioxidant activity of the ZANPs under high temperature in vitro. (**A**) Scavenging rate of anthocyanin extract and ZANPs on DPPH free radicals under high temperature treatment. (**B**) Percentage change in scavenging rate of A (Anthocyanin extract) and ZANPs. Within samples, values with different letters (a, b, c, d, e, f and g) are significantly different (*p* < 0.05).

**Figure 6 molecules-24-01421-f006:**
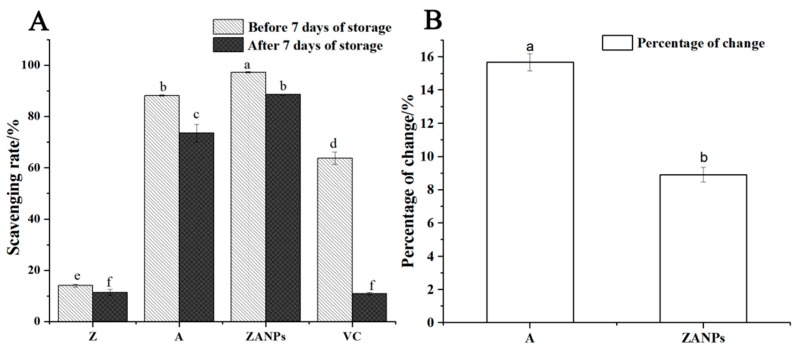
Stability and antioxidant activity of ZANPs after 7 days of storage in vitro. (**A**) Scavenging rate of anthocyanin extract and ZANPs on DPPH free radicals after 7 days of storage. (**B**) Percentage change in scavenging rate of A (Anthocyanin extract) and ZANPs. Within samples, values with different letters (a, b, c, d, e and f) are significantly different (*p* < 0.05).

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
