# Peer review of "Nanocrystallization of Anthocyanin Extract from Red-Fleshed Apple ′QN-5′ Improved Its Antioxidant Effect through Enhanced Stability and Activity under Stressful Conditions"

_molecules, 2019, doi:10.3390/molecules24071421_

Round 1
Reviewer 1 Report
Molecules-464224 - In this work, the authors quantified total phenols (Folin-Ciocalteu method), anthocyanins (ANTO; pH differential method) and antioxidant capacity (DPPH, .OH. and OH radicals) of red-flesh 'QN-5'' apple (peel and pulp), a variety developed by the authors. Ethanolic extracts from both samples were further nano-encapsulated (1:0, 1:0.1, 1:0.5 and 1:1) in zein (in ethanol; pH 11.0) by CaCl2-mediated coacervation. Size (200-500 nm), polydispersity, and antioxidant capacity of zein-ANTO nano-particles (ZANPs) were further characterized and their resistance to an alkaline treatment, to temperature and 7d storage were also evaluated.
In their article, the authors concluded that ZANPs made with a zein-to-ANTO ratio of 1: 0.5, increased the resistance to pH, temperature and storage of the original anthocyanin extract and that this type of molecular entrapment allowed ANTO controlled release at pH 1.2 and 7.4. The manuscript was prepared according to Molecules but this reviewer detected several issues in the experimental design and assay performance that reduces the quality of data presented. Some concerns are as follows:
A) Justification as to why an ANTO rich fruit with natural-pectin protection should be encapsulated in other covering material. Moreover, anthocyanins are very unstable molecules at basic (even at neutral pH) and ‘nanocrystals” are in fact ‘microcapsules’.
B) ANTO content was evaluated by the pH differential method that is OK for evaluating initial richness in raw materials (peel and pulp) but not after encapsulation, in vitro release, or in stability tests of the ANTO extract (by the way, they do not mentioned if it came from peel or pulp).
C) Basic calculations using data from section 4.7 indicate that the authors exposed 10, 50 and 100 mg of ANTO extract to 10 mg of zein in which case the covering material is so low for this purpose and do not explain their extremely high microencapsulation efficiency.
D) Antioxidant capacity (particularly DPPH) is so high, a fact related to an inconvenient assay (DPPH radical<<<< phenolic concentration) as evidenced in Figure 2 (same radical scavenging 10-50 mg.kg-1).
Author Response
Dear Editor and Reviewers:
First we would like to thank the editor and reviewers very much for the comments.
Reviewers' comments:
Reviewer #1: In this work, the authors quantified total phenols (Folin-Ciocalteu method), anthocyanins (ANTO; pH differential method) and antioxidant capacity (DPPH, .OH. and OH radicals) of red-flesh 'QN-5'' apple (peel and pulp), a variety developed by the authors. Ethanolic extracts from both samples were further nano-encapsulated (1:0, 1:0.1, 1:0.5 and 1:1) in zein (in ethanol; pH 11.0) by CaCl2-mediated coacervation. Size (200-500 nm), polydispersity, and antioxidant capacity of zein-ANTO nano-particles (ZANPs) were further characterized and their resistance to an alkaline treatment, to temperature and 7d storage were also evaluated.
In their article, the authors concluded that ZANPs made with a zein-to-ANTO ratio of 1: 0.5, increased the resistance to pH, temperature and storage of the original anthocyanin extract and that this type of molecular entrapment allowed ANTO controlled release at pH 1.2 and 7.4. The manuscript was prepared according to Molecules but this reviewer detected several issues in the experimental design and assay performance that reduces the quality of data presented. Some concerns are as follows:
Comment: 1. Justification as to why an ANTO rich fruit with natural-pectin protection should be encapsulated in other covering material. Moreover, anthocyanins are very unstable molecules at basic (even at neutral pH) and ‘nanocrystals” are in fact ‘microcapsules’.
Response: Thank the viewer very much for this comment. The materials in this article were anthocyanin extracts from peel and pulp of red flesh apple, and anhydrous ethanol were used as extraction agent. Natural-pectin that protects anthocyanin is insoluble in absolute ethanol and other organic solvents, so the anthocyanin extracts almost didn’t contain pectin. Anthocyanins are very unstable molecules, so, we used the zein as encapsulation for the anthocyanins to enhance its antioxidant ability and to alleviate cell damages by oxidative stress.
ZANPs may have similar properties to microcapsules, but the diameter of microcapsules is in the millimeter or micron level, while the diameter of ZANPs is in the range of 40-60 nm.
Comment: 2. ANTO content was evaluated by the pH differential method that is OK for evaluating initial richness in raw materials (peel and pulp) but not after encapsulation, in vitro release, or in stability tests of the ANTO extract (by the way, they do not mentioned if it came from peel or pulp).
Response: Thank the viewer very much for this comment. I'm very sorry, because of our negligence caused the error in writing. The determination of anthocyanin content in raw materials (peel and pulp) was evaluated by the pH differential method, after encapsulation and in vitro release, anthocyanin content was determined by UV-visible spectrophotometric method according to the reference (Arroyo-Maya, I.J.; McClements, D.J.; Biopolymer nanoparticles as potential delivery systems for anthocyanins: Fabrication and properties. Food Research International 2015, 69, 1-8.). It has been revised in line of 391#: “The anthocyanin content was determined by UV-visible spectrophotometric method [19]”.
In the tests of encapsulation, in vitro release, stability, the ANTO extract from pulp of red flesh apple was used as the materials because of its high anthocyanin content. It has been revised in line of 109#: The anthocyanin extract of pulp was allowed to prepare ZANPs because of its high anthocyanin content.
Comment: 3. Basic calculations using data from section 4.7 indicate that the authors exposed 10, 50 and 100 mg of ANTO extract to 10 mg of zein in which case the covering material is so low for this purpose and do not explain their extremely high microencapsulation efficiency.
Response: Thanks for the reviewer careful revision. I'm very sorry, because of our negligence caused the writing mistake. In section 4.7, 0.1,0.5 and 1.0 is the proportional of protein to anthocyanin, NOT concentration, the correct concentration is 0.01, 0.05, and 0.1 mg/mL. It has been revised in line of 364#: Then, the protein solution was incubated under magnetic agitation at 30 ℃ for 3 min. 100mL of anthocyanin extract which came from flesh (0.01, 0.05, and 0.1 mg/mL) were added dropwise into 100 mL zein ethanol solution respectively, while being mixed under continuous stirring (600 r/min) at 50 ℃.
Comment: 4. Antioxidant capacity (particularly DPPH) is so high, a fact related to an inconvenient assay (DPPH radical<<<< phenolic concentration) as evidenced in Figure 2 (same radical scavenging 10-50 mg.kg-1).
Response: Thank the viewer very much for this comment. In fact, we set more concentration gradients for anthocyanin antioxidant test. Our results also proved that anthocyanin had different scavenging rates on three free radicals, and the highest scavenging rate was to DPPH. At low anthocyanin concentration (0-10 mg.kg-1), DPPH scavenging rate increased significantly with the increase of anthocyanin concentration (Data was not shown in this article). When the concentration of anthocyanin was more than 10 mg.kg-1 (10-50 mg.kg-1), the DPPH scavenging rates did not increase significantly with the increase of anthocyanin concentration (Figure 2).
In this experiment, in order to compare the scavenging rates of anthocyanin on three free radicals conveniently, three same concentrations of 10, 30, 50 mg.kg-1 were used (Figure 2).
Reviewer 2 Report
Manuscript ID: molecules-464224
Title: Nanocrystallization of Anthocyanin Extract from Red-Fleshed Apple ‘QN-5’Improved Its Antioxidant Effect through Enhancing Stability and Activity under Stressful Conditions
The manuscript describes the synthesis and characterization of hybrid nanoparticles obtained by encapsulation of anthocyanin in zein protein, in comparison with anthocyanin extract. Generally, the article is well written, clear and concise.
1. I have few observations regarding the nanoparticles’ characterization. In Fig. 3 C-E the scale must be inserted. The magnification is also required.
2. How was calculated the mean particle size from TEM images? This information should be included in the manuscript.
3. All abbreviations must be explained in text before their first use.
4. Some words are linked (e.g. „1.6-3.0 timeslarger”, „100mlof”, etc.)
Author Response
Dear Editor and Reviewers:
First we would like to thank the editor and reviewers very much for the comments.
Reviewers' comments:
Reviewer #2: The manuscript describes the synthesis and characterization of hybrid nanoparticles obtained by encapsulation of anthocyanin in zein protein, in comparison with anthocyanin extract. Generally, the article is well written, clear and concise.
Comment: 1. I have few observations regarding the nanoparticles’ characterization. In Fig. 3 C-E the scale must be inserted. The magnification is also required.
Response: Thank the viewer very much for this suggestion. In Fig. 3 C-E the scale has been inserted.
Comment: 2. How was calculated the mean particle size from TEM images? This information should be included in the manuscript.
Response: Thanks for this comment. The range of particle size was calculated by the scale in the TEM image. It has been revised in line of 382#.
Comment: 3. All abbreviations must be explained in text before their first use.
Response: Thanks for this comment. It has been revised as the reviewer suggested. Please check the revised manuscript with highlight. In line of 26#: zein-anthocyanin nanoparticles (ZANPs); in line of 169#: transmission electron microscope, TEM
Comment: 4. Some words are linked (e.g. „1.6-3.0 timeslarger”, „100mlof”, etc.)
Response: It has been revised as the reviewer suggested. Please check the revised manuscript with highlight.
Round 2
Reviewer 1 Report
Molecules-464224-v2. The manuscript modified by the authors still has faults on justification, experimental design and interpretation of results. Here some already commented:
Comment 1 – It was misunderstood. What I meant is that you DID NOT justify why anthocyanins should be extracted and protected (nano-encapsulated) from this particular anthocyanin-rich apple variety. In addition, you mention that ZANPS size ranged from 40-60 nm when in in fact they are over 100 nm (Figure 3 A). The nanoscale commonly refers to particles below 100 nm and ZANPs are not.
Comment 2 – Ok with the new method although it needs to be more detailed (e.g. DMSO use, maximal wavelength and overall assay procedure) and it remains non-convenient to evaluate measure the influence of pH on anthocyanins.
As previously stated, anthocyanins are too sensitive to pH that modifies its “chromophore’ state and if the procedure used reestablishes pH. In fact, this sensitivity is convenient as a shelf life marker in some cases (Prietto et al.. 2017; https://doi.org/10.1016/j.lwt.2017.03.006). Arroyo-Maya & McClements (2015; https://doi.org/10.1016/j.foodres.2014.12.005) used commercial anthocyanin-rich encapsulated in whey protein isolate and beet pectin by thermal processing and electrostatic complexation and observed that at pH 3, their samples precipitated and so, reliable size and ANTO content measurements could not be made below pH 4 (you did it at 1.2). By the way, evaluations of pH-stability at pH 1.2 and 7.4 does not account as simulated “gastric” and “intestinal” conditions.
Comment 3 – OK with arguments, changes should be included in section 4.7 (no red changes appear in v2 of you manuscript).
Comment 4 – “At low anthocyanin concentration (0-10 mg.kg-1), DPPH scavenging rate increased significantly with the increase of anthocyanin concentration (Data was not shown in this article).” You better include it to make more evident a plausible “ZANPs’ antioxidant saturation rate”
After considering the above, you still consider re-submitting the manuscript, then you should also modify the following:
· What is VC. Explain the first time you mention it
· Improve resolution of graphs and indicate statistical differences with controls or other treatments when occur.
· Line 108-109 why did you use pulp instead of peel if the latter has three times more phenolic compounds? Discuss more all chemical differences between pulp and peel.
· Reduce significantly the abstract, but without sacrificing quantitative information and statistical differences. Include a brief description of why the anthocyanins must be protected from the pulp of this particular fruit.
· Zein alone has antioxidant capacity (https://onlinelibrary.wiley.com/doi/full/10.1111/1750-3841.12686). Include experimental data (ANTO content, antioxidant capacity) of unloaded (ANTO) zein-nanocapsules.
· Eliminate throughout the manuscript any argument/assumption of simulated (in vitro) digestion conditions if they were not properly evaluated as such (e.g. lines 94-95, section 2.5, Figure 7, etc. ).
Author Response
Response to reviewer’s comments V2
Dear Editor and Reviewers:
Thanks very much for the comments. We have revised and improved the whole article.
Reviewers' comments:
Molecules-464224-v2.The manuscript modified by the authors still has faults on justification, experimental design and interpretation of results. Here some already commented:
Comment 1–It was misunderstood. What I meant is that you DID NOT justify why anthocyanins should be extracted and protected (nano-encapsulated) from this particular anthocyanin-rich apple variety.In addition, you mention that ZANPS size ranged from 40-60 nm when in in fact they are over 100 nm (Figure 3 A). The nanoscale commonly refers to particles below 100 nm and ZANPs are not.
Response: The anthocyanins used for nano-encapsulation needed high concentration. In the anthocyanin encapsulation test, three different proportions of zein-to-anthocyanin (1:0.1, 1:0.5, 1:1.0) were set for characterizing the antioxidant properties of ZANPs, and the concentrations of anthocyanin extract were 0.01, 0.05, and 0.1 mg/mL, respectively. The anthocyanin content of the red flesh apple cultivar 'QN-5' is 361 mg·kg-1 (FW), and it can meet the requirement of anthocyanin concentration for encapsulation, other varieties with low anthocyanin content do not meet this requirement. Therefore, we used this particular anthocyanin-rich apple variety as material to extract anthocyanins for encapsulation experiment.
The particle size in Figure 3A was determined by DLS instrument. DLS measures the hydrodynamic diameter corresponding to the size of nanoparticle cluster and organic surfactant, whereas TEM detects boundaries of the individual magnetic nanoparticle. Therefore, the particle size measured by DLS was larger than that obtained by TEM [1]. It was mentioned at the end of the third paragraph in the discussion section of this article. DLS measurement has been added in Figure 3.
[1] Sarmphim, P.; Soontaranon, S.; Sirisathitkul, C.; Harding, P.; Kijamnajsuk, S.; Chayasombat, B,; Chingunpitak, J. FePt 3 nanosuspension synthesized from different precursors-a morphological comparison by SAXS, DLS and TEM. Bulletin of the Polish Academy of Sciences Technical Sciences 2017, 65.
Comment 2–Ok with the new method although it needs to be more detailed (e.g. DMSO use, maximal wavelength and overall assay procedure) and it remains non-convenient to evaluate measure the influence of pH on anthocyanins.
As previously stated, anthocyanins are too sensitive to pH that modifies its “chromophore’ state and if the procedure used reestablishes pH. In fact, this sensitivity is convenient as a shelf life marker in some cases (Prietto et al.. 2017; https://doi.org/10.1016/j.lwt.2017.03.006). Arroyo-Maya &McClements (2015; https://doi.org/10.1016/j.foodres.2014.12.005) used commercial anthocyanin-rich encapsulated in whey protein isolate and beet pectin by thermal processing and electrostatic complexation and observed that at pH 3, their samples precipitated and so, reliable size and ANTO content measurements could not be made below pH 4 (you did it at 1.2). By the way, evaluations of pH-stability at pH 1.2 and 7.4 does not account as simulated “gastric” and “intestinal” conditions.
Response: The new method with more detailed has been revised as the reviewer suggested.
After careful consideration, we have deleted throughout the manuscript any argument/assumption of simulated (in vitro) digestion conditions (e.g. section 2.5, Figure 7, etc. ).
Comment 3–OK with arguments, changes should be included in section 4.7 (no red changes appear in v2 of you manuscript).
Response: It has been revised as the reviewer suggested in section 4.7.
Comment 4–“At low anthocyanin concentration (0-10 mg.kg-1), DPPH scavenging rate increased significantly with the increase of anthocyanin concentration (Data was not shown in this article).” You better include it to make more evident a plausible “ZANPs’ antioxidant saturation rate”
Response: It has been revised as the reviewer suggested in section 2.2.
In order to make more evident a plausible “ZANPs’ antioxidant saturation rate”, we added a Figure S1. At low anthocyanin concentration (0-10 mg.kg-1), DPPH scavenging rate increased significantly with the increase of anthocyanin concentration.
Figure S1. Scavenging rate of anthocyanin extract on DPPH free radicals.
After considering the above, you still consider re-submitting the manuscript, then you should also modify the following:
1、What is VC. Explain the first time you mention it.
Response: It has been revised as the reviewer suggested.
2、Improve resolution of graphs and indicate statistical differences with controls or other treatments when occur.
Response: It has been revised as the reviewer suggested, Figure 1 and Figure 2 have been improved resolution.
3、Line 108-109 why did you use pulp instead of peel if the latter has three times more phenolic compounds? Discuss more all chemical differences between pulp and peel.
Response: In section 2.1, the differences of anthocyanin content and total phenol content between peel and pulp were compared and analyzed. Although the total phenol content in peel was higher than that in flesh, the anthocyanin content in peel was lower than that in flesh. The proportion of anthocyanin in total phenols in flesh was higher than that in peel. The nano-encapsulation test needed high anthocyanins concentration and content. So flesh with high anthocyanins content was suitable for the subsequent experiment. It has been revised as the reviewer suggested in section of discussion.
4、Reduce significantly the abstract, but without sacrificing quantitative information and statistical differences. Include a brief description of why the anthocyanins must be protected from the pulp of this particular fruit.
Response: We have reduced significantly the abstract as the reviewer suggested and it has been explained why the anthocyanins must be protected from the pulp of this particular fruit in abstract and introduction section.
5、Zein alone has antioxidant capacity (https://onlinelibrary.wiley.com /doi/full/10.1111 /1750-3841.12686). Include experimental data (ANTO content, antioxidant capacity) of unloaded (ANTO) zein-nanocapsules.
Response: Yes, zein alone has antioxidant capacity, it was shown in Figure 3B at zein-to-anthocyaninratios of 1:0. The data was antioxidant capacity of zein alone. Eencapsulation efficiency was zero without anthocyanin.
6、Eliminate throughout the manuscript any argument/assumption of simulated (in vitro) digestion conditions if they were not properly evaluated as such (e.g. lines 94-95, section 2.5, Figure 7, etc. ).
Response: It has been eliminated throughout the manuscript any argument/assumption of simulated as the reviewer suggested.
